# Unlocking the power of reticulocyte analysis: Advancing care for patients with abnormal haemoglobin variants at Eastern Regional Hospital, Koforidua

**Richard Vikpebah Duneeh** [1,2]*, **Israel Bedzina**[1,3], **Melody Yaaba Korkor Johnson**[1], **Francis Agyei Amponsah**[2], **Wina Ivy Ofori Boadu** [2], **Kenneth Ablordey**[1]

1 Department of Medical Laboratory Sciences, University of Health and Allied Sciences, Ho, Ghana, 2 Department of Medical Diagnostics, Kwame Nkrumah University of Science and Technology, Kumasi, Ghana, 3 Reinbee Medical Laboratory and Wellness Centre, Ho, Ghana

* rvduneeh@uhas.edu.gh

## Abstract

Reticulocytes are immature erythrocytes that have been released into the peripheral circulation after being extruded from the bone marrow. However, it has not been investigated whether these reticulocyte indicators are helpful in treating conditions linked to abnormal haemoglobin variants. In the current investigation, participants with abnormal haemoglobin variants were evaluated for the utility of reticulocyte parameters using a high-end haematology analyser. We conducted a prospective cross-sectional study among 217 participants with abnormal haemoglobin variants at the Eastern Regional Hospital, Koforidua. Statistical analyses were performed with IBM SPSS V.26.0 (Chicago, IL, USA) and R programming language version 4.2.3. A p-value of $< 0.05$ was considered statistically significant for all analyses. We found a significant association between age groups and anaemia severity (p = 0.016). Haemoglobin levels varied significantly among abnormal haemoglobin variants ($p < 0.05$), with the "SF" phenotype showing the lowest levels. Production defects accounted for 1.4% of cases, while maturation defects were present in 91.7%. An increase in low-fluorescence reticulocytes was linked to a 0.03 g/dL rise in haemoglobin (β = 0.03, $p < 0.05$), whereas increases in high- and medium-fluorescence reticulocytes led to declines of 0.069 g/dL and 0.052 g/dL, respectively ($p < 0.05$). Haemoglobin levels were significantly lower in participants with maturation abnormalities ($p < 0.01$). The study found relationships between the severity of anaemia, age, and haemoglobin phenotypes in individuals with abnormal haemoglobin variants. Given the predominance of maturation defects, management aimed at addressing ineffective erythropoiesis is necessary. The differences in how reticulocyte subpopulations affect haemoglobin levels imply that reticulocyte indices may be helpful biomarkers for monitoring and guiding therapeutic approaches.

**Data availability statement:** All relevant data is within the manuscript and its Supporting information files.

**Funding:** The author(s) received no specific funding for this work.

**Competing interests:** The authors have declared that no competing interests exist.

## Introduction

Haemoglobin (Hb) is a crucial component of red blood cells that are responsible for carrying oxygen throughout the body. However, various genetic mutations can lead to the production of abnormal haemoglobin variants, which can significantly impact the health and well-being of affected individuals. Collectively, haemoglobinopathies are sickle cell disease and thalassemia. These are characterized by chronic anaemia, recurrent infections, and an increased risk of complications such as organ damage and premature mortality [1–4].

The degree of inefficient erythropoiesis has been linked to the quantity of reticulocytes and their maturation levels in various haemoglobinopathies [5] Reticulocyte analysis has become a valuable diagnostic tool in the management of individuals with abnormal haemoglobin variants. Reticulocytes are immature red blood cells that are an essential component of the erythropoietic process. High end haematology analysers are able to estimate different subpopulations of these reticulocytes in the various phases of development. Reticulocytes are divided into low-fluorescence reticulocytes (LFR), medium-fluorescence reticulocytes (MFR) and high-fluorescence reticulocytes (HFR) in Sysmex analysers. These subtypes reflect the different phases of maturation based on their RNA content [6,7].

However, the usefulness of these reticulocyte indices in diagnosis and the treatment of diseases associated with abnormal haemoglobin variants has not been explored. This study provides a current understanding of reticulocyte analysis in the context of abnormal haemoglobin variants. The study discussed the principles of reticulocyte analysis in a high-end haematology analyser, its clinical utility, and the potential benefits of incorporating this diagnostic approach into the care of patients with these conditions. By highlighting the importance of reticulocyte analysis in advancing patient care, we hope to contribute to the development of more effective and personalized treatment strategies for individuals affected by abnormal haemoglobin variants.

## Methodology

### Study design

This study was a prospective facility-based cross-sectional study conducted among patients with abnormal haemoglobin variants at the Eastern Regional Hospital, Koforidua. The recruitment period of this study was three months, September to November, 2023.

### Study site

This study was carried out at the Eastern Regional Hospital of the New Juaben North Municipality. The hospital currently has a bed capacity of 356, and serves as a referral point for approximately 16 district hospitals in the Eastern Region. The Municipality lies between longitudes 1030 'west and 0030 'east' and latitudes 60 and 70 north. The Municipality shares common boundaries with East-Akim Municipal to the northeast, Akwapim District to the east and south and Suhum-Kraboa-Coaltar District to the east.

## Study population

Study participants were individuals with abnormal haemoglobin variants seeking care at Eastern Regional Hospital in Koforidua, Ghana, during the data collection period.

## Inclusion criteria

The study included all individuals with abnormal haemoglobin variants who accessed healthcare at Eastern Regional Hospital and consented to participate in this study over the study period.

## Exclusion criteria

The study excluded individuals with abnormal haemoglobin variants who were seriously ill and patients with abnormal haemoglobin variants who were on haematinic as well as those who declined to consent to participate in the study.

## Sample size

In this study, 217 participants were recruited. A convenience sampling method was used to select the 217 individuals with abnormal haemoglobin variants for this study.

## Sample and data collection

About 3–4 ml of whole blood from participants was collected aseptically into EDTA anticoagulated tube by the standard Venus sample collection method. The complete blood count with reticulocyte parameters was estimated for each patient using the Sysmex high-end fluorescence haematology analyser (XE-5000). This analyser operates on the principle of light scattering and fluorescence flow cytometry for the measurement of reticulocytes. Other patient information (sickling status and Hb phenotype) was retrieved from the Laboratory Information Management System (LIMS) and recorded in the data collection sheet.

## Definition of production defects and maturation defects

A production defect was defined as reticulocyte counts below 0.0225 x$10^6$/μL while maturation defects was defined as elevated high-fluorescence reticulocytes, together with decreased medium- and low-fluorescence reticulocytes. According to the fluorescence intensity that reflects the maturity of reticulocytes, (LFR: normal values between 81.0% and 96.4%; MFR: between 1.1% and 15.2%; HFR: between 0.03% and 3.95% of the total reticulocytes) [7].

## Classification of anaemia based on haemoglobin status

1) Mild anaemia: Hb 10–12 g/dL in females and to 13.5 g/dL in males.

2) Moderate anaemia: Hb 8 to 9.9 g/dL.

3) Severe anaemia: Hb 6.5 to 7.9 g/dL.

4) Life-threatening anaemia: Hb < 6.5 g/dL [8–10].

## Data handling and analysis

All data collected from participants were processed and confidentially saved on a password-protected computer accessible to only the researchers. Data was entered, cleaned, and coded using Microsoft Excel 2021. Statistical analyses were performed using the Statistical Package for Social Sciences (IBM-SPSS) Version 26.0 (Chicago IL, USA) and the R Programming Language version 4.2.3. Categorical variables are presented as frequencies and percentages. Parametric

continuous variables are presented as mean and standard deviation, while nonparametric continuous variables are presented as median and interquartile range after the normality test using the Kolmogorov–Smirnov test. A bar graph was used to illustrate the prevalence of production and maturation defects among patients with abnormal haemoglobin variants. The chi-square ($\chi^2$) test was used to assess factors associated with the severity of anaemia among study participants. Comparisons between study groups were calculated using the Kruskal–Wallis test followed by a post hoc test using the Bonferroni pairwise comparison test. Additionally, Spearman's correlation and a linear regression model were used to assess the association between reticulocyte indices and haemoglobin levels. A *P-value* of $< 0.05$ was considered statistically significant for all analyses.

### Ethical consideration

Ethical approval was obtained from the Research Ethics Committee of the University of Health and Allied Sciences with reference number UHAS-REC A.8 [49] 22–23. Permission from Management of the Eastern Regional Hospital was obtained for this study and the Helsinki declaration was observed as required. Written consent was obtained from adults' participants whiles assent was obtained from guardians and parents for children (1–17 years).

## Results

### Sociodemographic characteristics of patients with abnormal haemoglobin variants

The median age of the study participants was 7.00 years, with the majority within 1–5 years (40.1%), followed by 6–10 years (29.0%). Half of the study participants were males (50.2%) and the other half were females (49.8%). The predominant haemoglobin phenotypes were "S" (60.8%) and "SC" (34.6%). A significant number were also 'AS' (2.8%) and 'SF' (1.4%). Most of the participants had moderate anaemia (45.2%), some had severe anaemia (24.0%) and mild anaemia (19.8%), with 11.1% having severe life-threatening anaemia (Table 1).

### Reticulocyte characteristics of patients with abnormal haemoglobin variants

The median reticulocyte count and the relative reticulocyte count were 0.17 x$10^6$/µL (0.10–0.25) and 5.40% (2.65–8.57), respectively. Furthermore, the median fraction of immature reticulocytes and reticulocyte haemoglobin levels were 36.10% and 30.35%, respectively. The median of low, medium and high fluorescence levels were 63.70%, 17.80%, and 17.50%, respectively. Most of the participants had elevated absolute reticulocyte counts (60.8%), relative reticulocytes (76.5%), and immature reticulocyte fractions (94.5%). Furthermore, high medium (89.4%) and high (94.5%) fluorescence reticulocytes, alongside decreased low fluorescence (92.6%) and reticulocyte haemoglobin levels (67.1%) (Table 2).

### Sociodemographic factors associated with the severity of anaemia

Most of the participants aged from 1–5 years had mild anaemia (41.9%). However, most participants with severe life-threatening anaemia were aged from 20–51 years (37.5%) and 11–19 years (25.0%). Therefore, a significant association was found between the age group of the study participants and the severity of anaemia ($p = 0.016$) among the study participants.

More than half of the participants with mild (53.5%) and moderate (54.1%) anaemia were males, while most with severe anaemia (55.8%) and severe life-threatening anaemia (58.3%) were females. However, there was an insignificant association between gender and the severity of anaemia ($p = 0.529$) among study participants. Additionally, most patients with mild (53.5%) and moderate (60.2%) anaemia had 'S' haemoglobin phenotypes, followed by 'SC' haemoglobin phenotypes (39.5% and 36.7%, respectively). Similarly, most subjects with severe (65.4%) and severe life-threatening (66.7%) anaemia had 'S' haemoglobin phenotypes followed by 'SC' phenotypes (28.8% and 29.2%, respectively). Therefore,

**Table 1. Sociodemographic characteristics of patients with abnormal haemoglobin variants.**

| Variables | Frequency (n = 217) | Percentage (%) |
|---|---|---|
| **Age [Median (IQR)]** | 7.00 (4.00-12.00) | – |
| **Age Group (Years)** | | |
| 1-5 | 87 | 40.1 |
| 6-10 | 63 | 29.0 |
| 11-19 | 33 | 15.2 |
| 20-51 | 34 | 15.7 |
| **Gender** | | |
| Male | 109 | 50.2 |
| Female | 108 | 49.8 |
| **Haemoglobin Phenotype** | | |
| AS | 6 | 2.8 |
| AC | 1 | 0.5 |
| SC | 75 | 34.6 |
| SF | 3 | 1.4 |
| S | 132 | 60.8 |
| **Haemoglobin Level (g/dL) (π ± SD)** | 8.46 ± 1.68 | |
| **Anaemic Status** | | |
| Mild | 43 | 19.8 |
| Moderate | 98 | 45.2 |
| Severe | 52 | 24.0 |
| Severe life threatening | 24 | 11.1 |

haemoglobin phenotypes were not significantly associated with the severity of anaemia among study participants ($p = 0.381$) (Table 3).

## Comparison of haemoglobin levels between haemoglobin phenotypes

This study observed a significant difference in haemoglobin levels between the phenotypes ($p < 0.05$) of the study participants. In a post hoc pairwise comparison test by Bonferroni, patients with the haemoglobin phenotype of 'SF' had significantly lower haemoglobin levels than those with the haemoglobin phenotypes of 'S', 'SC', 'AS' and 'AC' ($p < 0.05$) (Fig 1).

## Prevalence of defects in reticulocyte production and maturation among patients with abnormal haemoglobin variants

This study found that the majority of patients with abnormal haemoglobin variants had maturation defects, (91.7%), while the prevalence of production defects among patients with abnormal haemoglobin variants was 1.4% (Fig 2).

## Association between reticulocyte production indices and haemoglobin level

Figs 3 and 4 show the Spearman correlations and the linear regression model of the effect of reticulocyte indices on haemoglobin levels among patients with abnormal haemoglobin variants. There were significant weak negative correlations between reticulocyte count ($r = -0.152$, $p = 0.026$), reticulocyte percentage ($r = -0.300$, $p < 0.001$), immature reticulocyte fraction ($r = -0.197$, $p = 0.004$), and haemoglobin levels among patients with abnormal haemoglobin variants (Fig 3A–C). However, there was an insignificant weak positive correlation between reticulocyte haemoglobin levels ($r = 0.029$, $p = 0.675$) and haemoglobin levels among patients with abnormal haemoglobin variants (Fig 3D).

**Table 2. Reticulocyte characteristics of study participants.**

| Variables | Frequency (n = 217) | Percentage (%) |
|---|---|---|
| **Reticulocyte Count ($10^6$/uL)** | | |
| Normal | 80 | 36.9 |
| Decreased | 5 | 2.3 |
| Elevated | 132 | 60.8 |
| **Reticulocyte (%)** | | |
| Normal | 48 | 22.1 |
| Decreased | 3 | 1.4 |
| Elevated | 166 | 76.5 |
| **Immature Reticulocytes Fraction (%)** | | |
| Normal | 12 | 5.5 |
| Elevated | 205 | 94.5 |
| **Reticulocyte Haemoglobin (pg)** | | |
| Normal | 64 | 29.6 |
| Decreased | 145 | 67.1 |
| Elevated | 7 | 3.2 |
| **Low Fluorescence Reticulocytes (%)** | | |
| Normal | 16 | 7.4 |
| Decreased | 201 | 92.6 |
| **Medium Fluorescence Reticulocytes (%)** | | |
| Normal | 23 | 10.6 |
| Elevated | 194 | 89.4 |
| **High Fluorescence Reticulocytes (%)** | | |
| Normal | 12 | 5.5 |
| Elevated | 205 | 94.5 |
| **Variables** | **Median (range)** | |
| **Reticulocyte Count ($10^6$/uL)** | 0.17 (0.10-0.25) | – |
| **Reticulocyte (%)** | 5.40 (2.65-8.57) | – |
| **Immature Reticulocytes Fraction (%)** | 36.10 (27.30-42.80) | – |
| **Reticulocyte Haemoglobin (pg)** | 30.35 (25.80-33.28) | – |
| **Low Fluorescence Reticulocytes (%)** | 63.70 (57.00-72.35) | – |
| **Medium Fluorescence Reticulocytes (%)** | 17.80 (15.10-20.00) | – |
| **High Fluorescence Reticulocytes (%)** | 17.50 (11.20-22.55) | – |

In a linear regression prediction model, an increase in 1 x$10^6$μl of reticulocyte counts was significantly associated with a 0.12 g/dL decrease in haemoglobin levels ($\beta$ = -0.12, $p < 0.05$) among patients with abnormal haemoglobin variants (Fig 3A). Furthermore, a percentage increase in reticulocyte counts among patients with abnormal haemoglobin variants resulted in a 0.08 g/ dL decrease in haemoglobin levels ($\beta$ = -0.08, $p < 0.05$) (Fig 3B). Similarly, a percentage increase in the immature reticulocyte fraction was associated with a significant decrease of 0.030 g/ dl in haemoglobin levels ($\beta$ = -0.0.03, $p < 0.05$) among patients with abnormal haemoglobin variants (Fig 3C). In contrast, a 1 pg increase in reticulocyte hae-moglobin levels was associated with a 0.016 g/dL increase in haemoglobin levels ($\beta$ = 0.016, $p < 0.05$) among patients with abnormal haemoglobin variants (Fig 3D).

## Association between reticulocyte maturation-dependent indices and haemoglobin level

This study found a significant weak positive correlation between low fluorescence reticulocytes ($r$ = 0.171, $p$ = 0.012) and haemoglobin levels among patients with abnormal haemoglobin variants (Fig 4A). However, there was a significant weak

**Table 3. Sociodemographic factors associated with the severity of anaemia among patients with abnormal haemoglobin variants.**

| Variables | Total (n=217) | Anaemia Severity | | | | p value |
| --- | --- | --- | --- | --- | --- | --- |
| | | Mild (n=43) | Moderate (n=98) | Severe (n=52) | Severe Life Threatening (n=24) | |
| **Age Group (Years)** | | | | | | **0.016** |
| 1-5 | 87 (40.1) | 18 (41.9) | 45 (45.9) | 20 (38.5) | 4 (16.7) | |
| 6-10 | 63 (29.0) | 11 (25.6) | 26 (26.5) | 21 (40.4) | 5 (20.8) | |
| 11-19 | 33 (15.2) | 5 (11.6) | 16 (16.3) | 6 (11.5) | 6 (25.0) | |
| 20-51 | 34 (15.7) | 9 (20.9) | 11 (11.2) | 5 (9.6) | 9 (37.5) | |
| **Gender** | | | | | | 0.529 |
| Male | 109 (50.2) | 23 (53.5) | 53 (54.1) | 23 (44.2) | 10 (41.7) | |
| Female | 108 (49.8) | 20 (46.5) | 45 (45.9) | 29 (55.8) | 14 (58.3) | |
| **Haemoglobin Phenotype** | | | | | | 0.381 |
| AS | 6 (2.8) | 2 (4.7) | 3 (3.1) | 1 (1.9) | 0 (0.0) | |
| AC | 1 (0.5) | 1 (2.3) | 0 (0.0) | 0 (0.0) | 0 (0.0) | |
| SC | 75 (34.6) | 17 (39.5) | 36 (36.7) | 15 (28.8) | 7 (29.2) | |
| SF | 3 (1.4) | 0 (0.0) | 0 (0.0) | 2 (3.8) | 1 (4.2) | |
| S | 132(60.8) | 23 (53.5) | 59 (60.2) | 34 (65.4) | 16 (66.7) | |

P value is significant at $p < 0.05$.

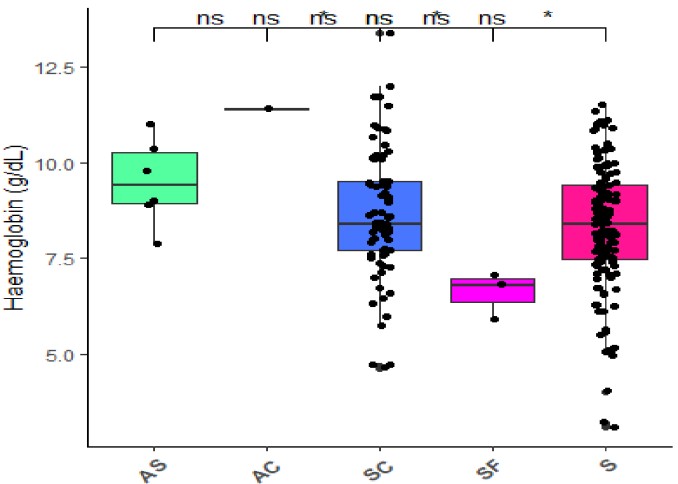

**Fig 1. Comparison of haemoglobin levels between different haemoglobin phenotypes among patients with abnormal haemoglobin variants.**
Note: the $p$ values between groups were calculated using the Kruskal-Wallis test, and the post hoc test was using the Bonferroni pairwise comparison test, ns: not significant at $p > 0.05$, significant at *: $p < 0.05$, **: $p < 0.01$, ***: $p < 0.001$.

negative correlation between high fluorescence reticulocytes ($r=-0.289$, $p<0.001$) and haemoglobin levels and an insignificant weak negative correlation between medium fluorescence reticulocytes ($r=-0.122$, $p=0.075$) and haemoglobin levels among patients with abnormal haemoglobin variants (Fig 4B, C).

In a linear regression prediction model, a percentage increase in low-fluorescence reticulocytes was significantly associated with an increase of 0.03 g/ dL in haemoglobin levels ($\beta=0.03$, $p<0.05$) among patients with abnormal haemoglobin variants (Fig 4A). However, a percentage increase in high-fluorescence reticulocytes ($\beta=-0.069$, $p<0.05$) and medium-fluorescence reticulocytes ($\beta=-0.052$, $p<0.05$) resulted in decreases of 0.069 g/dL and 0.052 g/dL, respectively, in haemoglobin levels among patients with abnormal haemoglobin variants (Fig 4B, C).

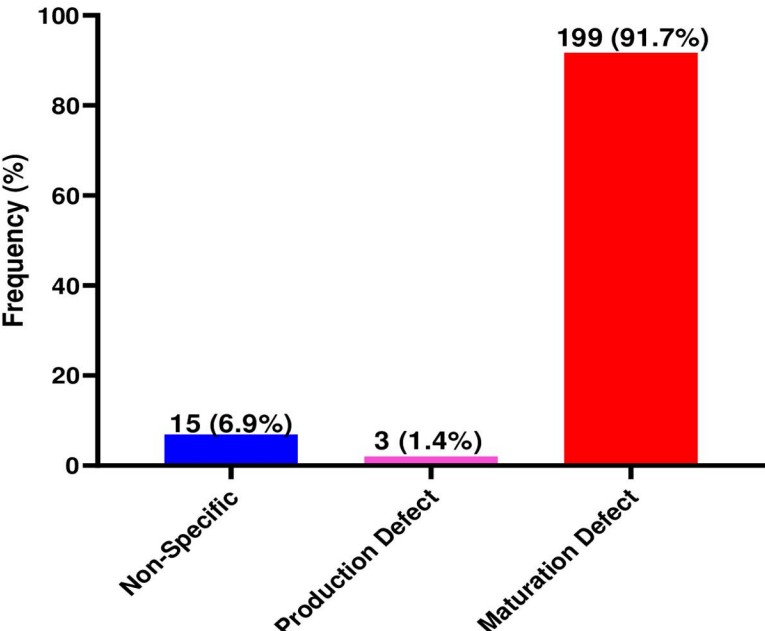

**Fig 2. Prevalence of reticulocyte production and maturation defects among patients with abnormal haemoglobin variants.**

## Comparison between haemoglobin, reticulocyte haemoglobin, and defective statuses

Fig 5 shows a box and violin diagram of the comparison of haemoglobin levels and reticulocyte haemoglobin levels between defective status among patients with abnormal haemoglobin variants. This study found that patients with maturation defects had significantly lower levels of haemoglobin ($p < 0.01$) compared to those with nonspecific causes of anaemia (Fig 5A).

However, the reticulocyte haemoglobin level was not significantly different between the defective state of patients with abnormal haemoglobin variants ($p > 0.05$) (Fig 5B).

## Discussion

This study found a significant association between age group and the severity of anaemia among patients with abnormal haemoglobin variants. The trend in the result shows that increasing age was a predictor of experiencing more severe anaemia among patients with abnormal haemoglobin variants. This is consistent with the study by Simbauranga *et al.'s* in Tanzania, who found that increasing age was associated with severe anaemia [11].

The cumulative effect of the condition over time can explain the significant association between age groups and the severity of anaemia in people with aberrant haemoglobin mutations. Patients are more likely as they age to experience prolonged and recurrent haemolytic episodes, which can deplete essential resources such as iron stores that are essential for the synthesis of red blood cells. Additionally, chronic organ damage, particularly the liver, kidneys, and spleen, makes it more difficult for the body to regulate erythropoiesis. Furthermore, older individuals often have higher rates of complications of haemoglobin problems, such as infections and Vaso-occlusive crises, exacerbated by the severity of anaemia [12–16].

In addition, the haemoglobin level was significantly different between different haemoglobin phenotypes. Despite being carriers, it was found that 2.8% of individuals with the 'AS' genotype were receiving treatment as individuals with abnormal haemoglobin variants. Could this be as a result of the AS patient having predominant haemoglobin S? A post hoc pairwise comparison test found that patients with the haemoglobin phenotype of 'SF' had significantly lower haemoglobin levels

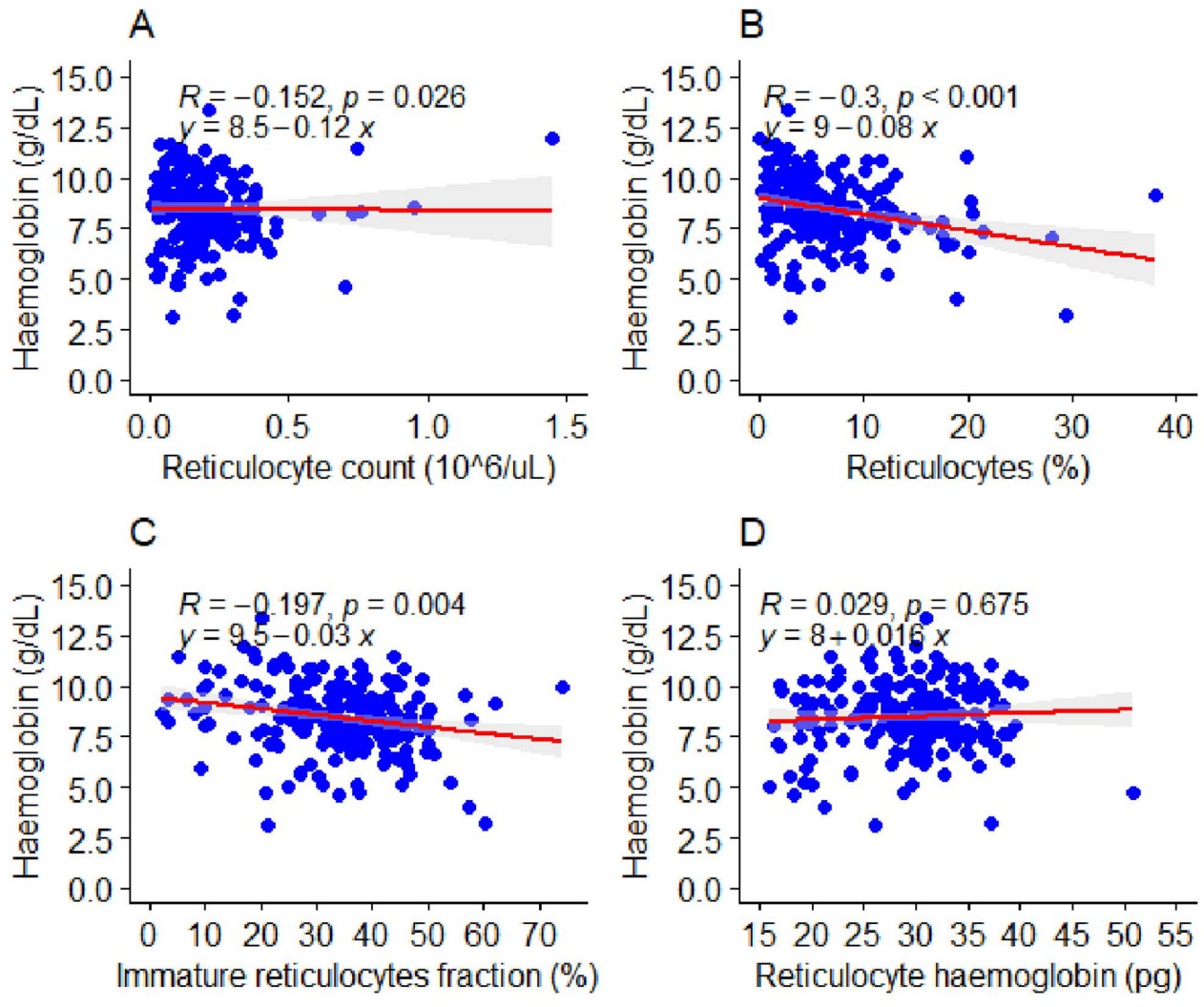

**Fig 3. Association between reticulocyte production indices and haemoglobin levels among patients with abnormal haemoglobin variants.**

than those with the haemoglobin phenotypes of 'S', 'SC', 'AS' and 'AC'. The reason for this striking finding was not known in this study hence further study is required. Contrary to this study, Simbauranga *et al.* [11] reported that haemoglobin 'S' and 'AS' were significantly associated with severe anaemia. Rodrigo *et al.* [17] in Sri Lankan adolescents also reported that haemoglobin variants are significantly associated with haemoglobin levels. The high incidence of mild to severe life-threatening anaemia could be explained by chronic haemolysis or hyper haemolytic crisis, folate or iron deficit caused by increased folate use and increased iron loss in urine, depression of erythropoiesis (aplastic crisis) and sequestration crisis among patients with abnormal haemoglobin variants [18].

The prevalence of maturation defects and production defects was 91.7% and 1.4%, respectively. Interestingly, most patients with abnormal haemoglobin variants have been found to experience haemolytic episodes [18] and infections [19]. This leads to increased anaemia and haematopoiesis to supplement the metabolic rates of the body. Reticulocytes, a marker of bone marrow haematopoietic activity, are typically elevated in haemolysis as well as in other pathological and physiological circumstances [20]. Enucleation of mature erythroblasts in the bone marrow results in the production

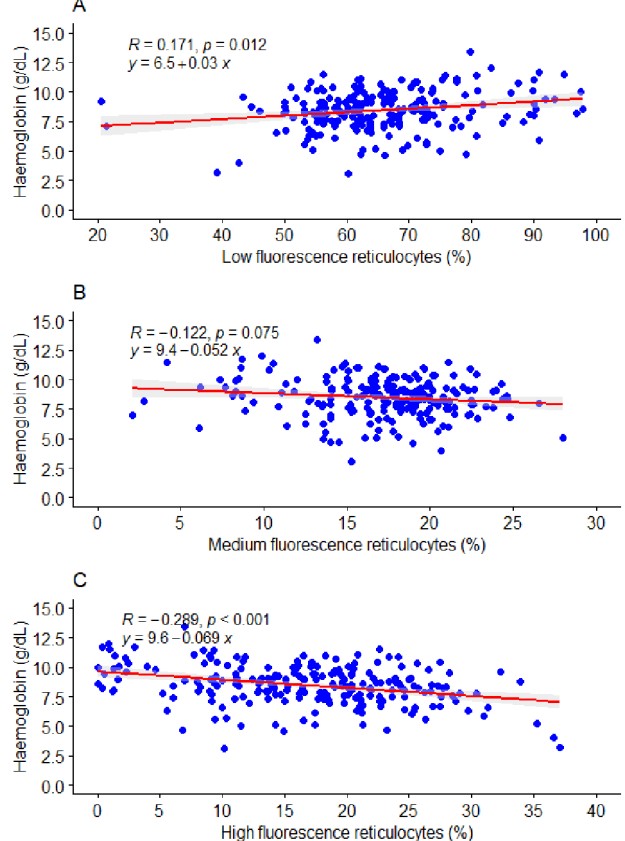

**Fig 4. Association between reticulocyte maturation-dependent indices and haemoglobin levels among patients with abnormal haemoglobin variants.** Note: R: The Spearman correlation value, y = m*x + c *, represents the linear regression equation where the variables change.

of reticulocytes. Reticulocytes are expelled from the bone marrow after only a short time, when they shed any remaining RNA and develop into erythrocytes. Under homeostatic circumstances, both the bone marrow compartment and the circulation require 2–3 days for most reticulocytes to mature [21,22]. The high prevalence of maturation defects in the current study shows high immature reticulocyte stages among patients with abnormal haemoglobin variants due to chronic haemolytic episodes. Nutritional deficiencies, particularly iron, vitamin B12, or folate deficiencies, may impair erythropoiesis and lead to altered reticulocyte counts. Similarly, recent blood transfusions can suppress reticulocyte production by alleviating anaemia and reducing erythropoietin stimulation. These unmeasured variables, though beyond the scope of this study, are critical considerations for interpreting reticulocyte data and warrant further investigation in future studies. The findings of this study call for interventions in the treatment of patients with abnormal haemoglobin variants that should be geared toward haemolytic episodes.

This study showed that an increase in $1 \times 10^6 \mu$l of reticulocyte counts was significantly associated with a 0.12 g/dL decrease in haemoglobin levels among patients with abnormal haemoglobin variants. Furthermore, a percentage increase in reticulocyte counts and immature reticulocyte fraction resulted in significant decreases of 0.08 g/dL and 0.030 g/dL, respectively, in haemoglobin levels among patients with abnormal haemoglobin variants. This confirms that patients with sickle cell disease are prone to chronic haemolytic episodes resulting in anaemia [23]. While transfusion provides a short-term solution for severe haemolysis, the findings of this study further show that there is also a compensatory mechanism by increasing haemopoiesis, which may lead to reticulocytosis. A 1 pg increase in reticulocyte haemoglobin

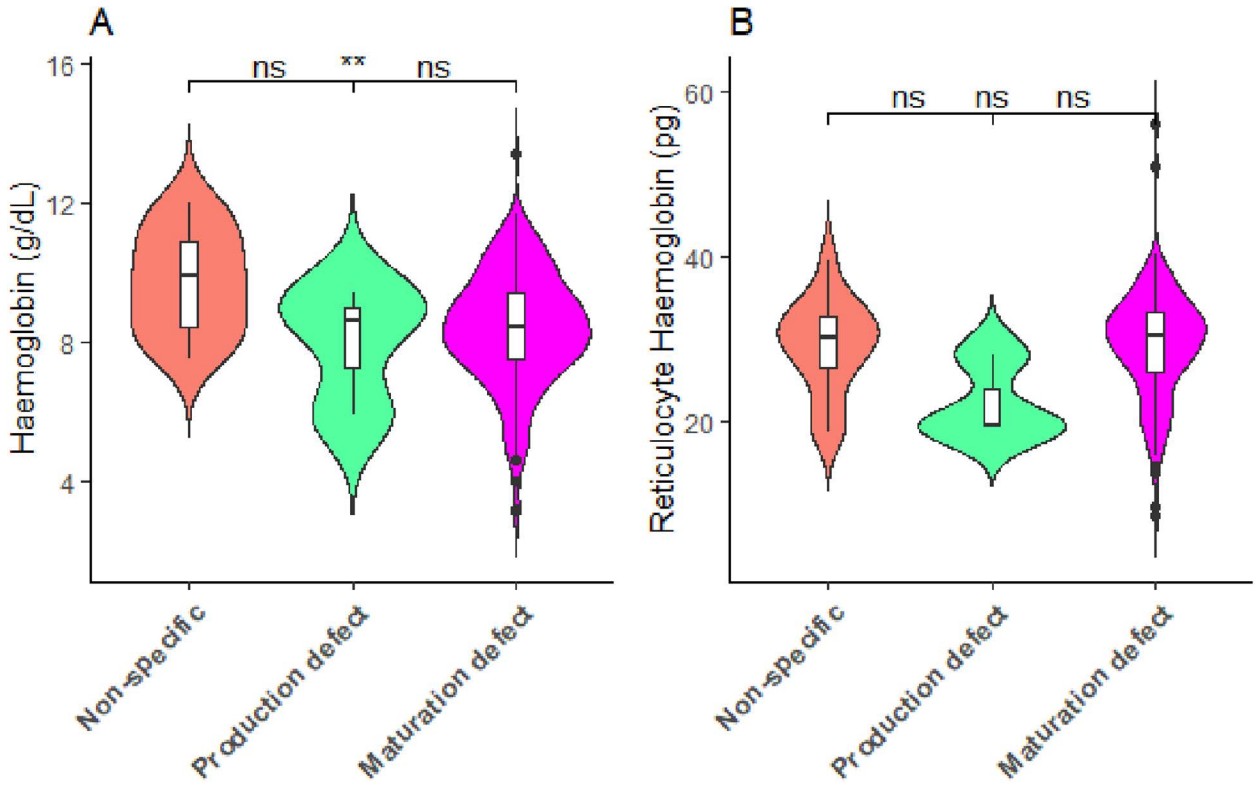

**Fig 5. Comparison between haemoglobin and reticulocyte haemoglobin levels and defect status among patients with abnormal haemoglobin variants.** Note: the *p* values between groups were calculated using the Kruskal-Wallis test, and the post hoc test was using the Bonferroni pairwise comparison test, ns: not significant at $p > 0.05$, *: $p < 0.05$, **: $p < 0.01$, ***: $p < 0.001$.

levels is associated with a 0.016 g/dL increase in haemoglobin levels, which can also be an indicator of possible short-term anaemia crisis relief among patients with abnormal haemoglobin variants. Furthermore, a percentage increase in low-fluorescence reticulocytes was significantly associated with an increase of 0.03 g/dL increase in haemoglobin levels among patients with abnormal haemoglobin variants, while a percentage increase in high-fluorescence reticulocytes and medium-fluorescence reticulocytes resulted in decreases of 0.069 g/dL and 0.052 g/dL decreases, respectively, in haemoglobin levels among patients with abnormal haemoglobin variants. Low-fluorescence reticulocytes are more mature cells with minimal RNA content, reflecting successful red blood cell maturation. An increase in Low-fluorescence reticulocytes indicates that the bone marrow is efficiently producing and maturing red blood cells, which supports higher haemoglobin levels.

In contrast, medium- and high-fluorescence reticulocytes have higher RNA content, signifying incomplete maturation. An increase in these immature reticulocytes typically indicates stress erythropoiesis, where the bone marrow releases cells prematurely due to conditions like chronic anaemia or dysfunction. This inefficiency in the maturation process results in fewer functional red blood cells, contributing to lower haemoglobin levels. These findings highlight the possible usefulness of fluorescence reticulocytes as a diagnostic approach to establish production and maturation defects among patients with abnormal haemoglobin variants.

This study found that patients with maturation defects had significantly lower levels of haemoglobin compared to those with nonspecific causes of anaemia. However, the level of reticulocyte haemoglobin was not significantly different between the defective status of patients with abnormal haemoglobin level. Maturation defects are measured by high-fluorescence

reticulocytes due to elevated immature reticulocytes as a result of haemolytic episodes in patients with SCD. This could be the possible decrease in haemoglobin levels in patients with SCD with maturation defects compared to those with nonspecific causes of anaemia. This study also acknowledges the lack of control group as a limitation. Future research should include a control group comprising individuals with normal haemoglobin variants to enable comparisons and provide context for the findings. This approach will enhance the generalizability and interpretability of the results. Fluorescence reticulocytes offer accurate way to evaluate bone marrow activity, which is useful for both diagnosing and tracking diseases such anaemia, bone marrow suppression, and treatment response. These may help to personalize interventions, such as modifying erythropoiesis-stimulating drugs in patients with renal failure or assessing recovery in patients who have received chemotherapy.

## Conclusions

The findings highlight the complex relationship between age, haemoglobin phenotypes, and the severity of anaemia in individuals with abnormal haemoglobin variants. The predominance of maturation defects underscores the need to focus on interventions that address ineffective erythropoiesis in this population. Additionally, the differential impact of reticulocyte subpopulations on haemoglobin levels suggests that reticulocyte indices could serve as valuable biomarkers for monitoring disease progression and tailoring treatment strategies. These insights emphasize the importance of comprehensive reticulocyte analysis in improving the management and outcomes of patients with abnormal haemoglobin variants.

## Limitations of this study

The study was carried out in only one facility, and so the findings may not be applicable to other settings. The study could not include confounding factors such nutritional status, transfusion history and infection history. Also, there was a lack of control group which limited our ability to compare our study to individuals with normal haemoglobin variants

## Recommendations

These findings suggest the use of fluorescence reticulocytes as a diagnostic marker to identify production and maturation defects in patients with abnormal haemoglobin variants. To enhance the diagnostic potential of fluorescence reticulocytes in the management of these patients, more longitudinal studies are necessary. Additionally, this study should be multi-centred to validate the results. Future studies should include control group to compare individuals with normal haemoglobin to those with abnormal haemoglobin as well as considering confounders such as transfusion history, infection history and nutritional status. We also encourage future studies to explore the role of reticulocytes in therapeutic monitoring for haemoglobinopathies.

## Supporting information

**S1 File. Anonymised retics data.**
(ZIP)

## Acknowledgments

The authors acknowledge the head and staff of the haematology department of the Eastern Regional Hospital, Koforidua. Ghana.

## Author contributions

**Conceptualization:** Richard Vikpebah Duneeh, Israel Bedzina, Melody Yaaba Korkor Johnson, Francis Agyei Amponsah, Wina Ivy Ofori Boadu, Kenneth Ablordey.

**Formal analysis:** Francis Agyei Amponsah, Kenneth Ablordey.

**Investigation:** Richard Vikpebah Duneeh, Melody Yaaba Korkor Johnson, Wina Ivy Ofori Boadu, Kenneth Ablordey.

**Methodology:** Richard Vikpebah Duneeh, Israel Bedzina, Melody Yaaba Korkor Johnson, Kenneth Ablordey.

**Project administration:** Richard Vikpebah Duneeh, Israel Bedzina, Francis Agyei Amponsah.

**Resources:** Richard Vikpebah Duneeh, Israel Bedzina, Melody Yaaba Korkor Johnson, Francis Agyei Amponsah, Wina Ivy Ofori Boadu, Kenneth Ablordey.

**Writing – original draft:** Israel Bedzina, Melody Yaaba Korkor Johnson, Francis Agyei Amponsah.

**Writing – review & editing:** Richard Vikpebah Duneeh, Israel Bedzina, Francis Agyei Amponsah, Wina Ivy Ofori Boadu.

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
