## [Decision Letter · Decision Letter 0]

19 Nov 2024

PONE-D-24-30332Unlocking the Power of Reticulocyte Analysis: Advancing care for patients with abnormal haemoglobin variants at Eastern Regional Hospital, KoforiduaPLOS ONE

Dear Dr. Duneeh,

Thank you for submitting your manuscript to PLOS ONE. After careful consideration, we feel that it has merit but does not fully meet PLOS ONE’s publication criteria as it currently stands. Therefore, we invite you to submit a revised version of the manuscript that addresses the points raised during the review process.

We look forward to receiving your revised manuscript.

Kind regards,

Ochuwa Adiketu Babah, M.Sc.PH (Epidemiology), FWACS, FMCOG

Academic Editor

PLOS ONE

**Journal Requirements:**

**Additional Editor Comments:**

Dear Authors,

Please address all comments made by the reviewers and submit for re-assessment.

Best regards,

Dr Ochuwa A. Babah

Reviewers' comments:

Reviewer's Responses to Questions

**Comments to the Author**

1. Is the manuscript technically sound, and do the data support the conclusions?

Reviewer #1: Yes

Reviewer #2: Yes

Reviewer #3: Yes

2. Has the statistical analysis been performed appropriately and rigorously? 

Reviewer #1: Yes

Reviewer #2: Yes

Reviewer #3: Yes

3. Have the authors made all data underlying the findings in their manuscript fully available?

Reviewer #1: No

Reviewer #2: Yes

Reviewer #3: Yes

4. Is the manuscript presented in an intelligible fashion and written in standard English?

Reviewer #1: Yes

Reviewer #2: Yes

Reviewer #3: Yes

5. Review Comments to the Author

**Reviewer #1:**  Study site Page 10; Line 85:

Suggestion: The description of the study site is too long. May be cut short.

Sample and Data Collection Page 12; Line 116:

Written consent was obtained from adults’ participants whiles assent was obtained from guardians and parents

Suggestion: Spell check. Assent is the a child's affirmative agreement to participate in research, and is taken from children between 7-17 years; when it is the agreement from the parent/guardian it is called as consent from the parent/guardian.

Definition of production defects and maturation defects Page- 12

Suggestion: What is the reference for these definitions? Is there any cut off for the fluorescence baed on which reticulocytes are categorised into high, medium and low?

Results  page 13; Lines 149-150

The median age of the study participants was 7.00 years,.... Half were men (50.2%) and half were women 151 (49.8%).

Suggestion: Better to mention the range along with median age. Mention as males and females rather than men and women as >70% were children.

Table 1page 13

Haemoglobin phenotype -'S".

Suggestion: What is Hb 'S' phenotype? Is it homozygous sickle? Then it is better to mention 'SS' phenotype

Alignment of the variables in the table to be corrected

Reticulocyte characteristics of patients with abnormal haemoglobin variantsPage 14; Lines 159-

The median reticulocyte count and the absolute reticulocyte count were 0.71 x10^6/μL and 5.40%, respectively.

Suggestions: How can the absolute count be in percentage? Always mention range along with median.

Lines 161-162

The median low, medium and high fluorescence levels were 63.70%, 17.80%, and 17.50%, respectively.

Suggestion: Is it the median low, medium and high fluorescence reticulocyte %?

Lines 163-164: Most

163 of the participants had elevated reticulocyte counts (60.8%), reticulocytes (76.5%), and 164 immature reticulocyte fractions (94.5%).

Suggestion: Sentence is not clear. How are Reticulocyte count and reticulocytes different here?

Lines 164-166:Furthermore, most had elevated medium (89.4%) and high (94.5%) fluorescence reticulocytes. However, most had decreased low fluorescence (92.6%) and reticulocyte haemoglobin levels (67.1%).

Suggestion: Rephrase the sentence. The message conveyed is not clear.

Table  Pages 14, 15

When you are mentioning absolute Reticulocyte count, Is there any significance for the reticulocyte %? Better index would be the reticulocyte index or reticulocyte production index.

The last portion in the table 2, can be incorporated in the upper part of the table only.

Eg;

Table 2: Reticulocyte characteristics of study participants (N=217)

Variable Median (range)/N(%)

Reticulocyte Count (106/uL) 0.17 (0.10-0.25)

Normal 16 (7.4)

Sociodemographic factors associated with the severity of anaemia page 15

Lines- 177-

Additionally, most patients with mild (53.5%) and moderate (60.2%) anaemia had 'S' haemoglobin phenotypes, followed by 'SC' haemoglobin phenotypes (39.5% and 36.7%, respectively). Similarly, most subjects with severe (65.4%) and severe life-threatening (66.7%) anaemia had 'SS' haemoglobin phenotypes followed by 'SC' phenotypes (28.8% and 29.2%, respectively).

Suggestion: Is Haemoglobin 'S' and 'SS' the same? Better to use the same short forms

DiscussionPage 21

Contrary to this study, (8) reported that SS and AS were significantly associated with severe anaemia. (14) in Sri Lankan adolescents also reported that haemoglobin variants are significantly associated with haemoglobin levels.

Suggestion: Please mention the study/author details; not just reference number.

Lines 297-

Interestingly, most patients with abnormal haemoglobin variants have been found to experience haemolytic episodes Patel et al., (2021) and infections Alkot et al., (2018). This leads to increased anaemia and haematopoiesis to supplement the metabolic rates of the body.

Suggestion: Sentences need rephrasing. Do not convey the message. Use same pattern of reference through out the manuscript.

The result and discussion parts are too long and confusing the reader. When all the findings are mentioned in the tables, mention only the relevant findings in the text part of the result.

The sentences in most places are incomplete and do not convey the clear message.

The patient selection, characteristics, inclusion and exclusion criteria are not clearly mentioned.

How many were transfusion dependent/symptomatic?

If transfusion dependent, what was the interval between transfusion and blood analysis for study in these patients?

Was the iron study done?

Was any patient on iron chelation?

Was there splenomegaly?

Overall this is an interesting study. The result and discussion part have to be done with more clarity.

**Reviewer #2: ** The manuscript has been well written but however, please go through my comments and make the necessary corrections suggested. I am commending you again for a good work done. I hope this continues and i get to review more papers from you.

**Reviewer #3:**  Abstract contains repeated statements which will need to be corrected.

Conclusion in abstract and the conclusion section of the manuscript should be re-written to clearly state findings and implications without necessarily repeating the results segment.

Statement on written consent and assent was repeated under sample collection and ethical approval. Several other repeated statements in the manuscript should be avoided.

"The median age of the study participants was 7.00 years, with the majority within 1-5 years (40.1%), followed by 6-10 years (29.0%). Half were men (50.2%) and half were women (49.8%)." - It is better to use males and females especially as the mean age of participants was just 7 years.

The levels of HB used to define and categorize anaemia should be stated either be in text or in the table especially since the age range of the study participants cut across both paediatric and adult age groups.

Figures presented do not adequately explain the findings described in text for example, the association between reticulocyte production indices and haemoglobin level and the association between reticulocyte maturation-dependent indices and haemoglobin levels among patients with abnormal haemoglobin variants would have been better presented in tables.

Referencing in the discussion section does not follow acceptable methods. Some referenced authors were not named in text while some that were named did not have the reference number indicated. (Line 289, 290 and 298 are examples)

The authors did not explain why the increase in low-fluorescence reticulocytes is associated with an increase in haemoglobin levels while increase in medium and high-fluorescence reticulocytes had an opposite effect.

Why were Haemoglobin A individuals not included in the study as controls? The findings in these individuals versus those with abnormal haemoglobin variants may have helped explain some of the findings not well addressed in the discussion section.

6. PLOS authors have the option to publish the peer review history of their article (what does this mean? ). If published, this will include your full peer review and any attached files.

**Do you want your identity to be public for this peer review?** For information about this choice, including consent withdrawal, please see our Privacy Policy .

Reviewer #1: No

Reviewer #2: **Yes: ** Juliana Aggrey

Reviewer #3: No

---

## [Author Response · Author response to Decision Letter 1]

12 Dec 2024

Reviewer #1:

Comment: Study site Page 10; Line 85:

Suggestion: The description of the study site is too long. May be cut short.

Response: Authors have summarised study site. See line 87-92.

Comment: Sample and Data Collection Page 12; Line 116:

Written consent was obtained from adults’ participants whiles assent was obtained from guardians and parents

Suggestion: Spell check. Assent is the a child's affirmative agreement to participate in research, and is taken from children between 7-17 years; when it is the agreement from the parent/guardian it is called as consent from the parent/guardian.

Response: Authors have revised the written consent accordingly. See line 145-146.

Comment: Definition of production defects and maturation defects Page- 12

Suggestion: What is the reference for these definitions? Is there any cut off for the fluorescence baed on which reticulocytes are categorised into high, medium and low?

Response: Authors have defined and added reference for these definitions. See line 116-120.

Results  page 13; Lines 149-150

Comment: The median age of the study participants was 7.00 years,.... Half were men (50.2%) and half were women 151 (49.8%).

Suggestion: Better to mention the range along with median age. Mention as males and females rather than men and women as >70% were children.

Response: We appreciate your input on the range, however our IQR was the best considering the dataset we had. Second option has been revised accordingly. See line 150-155.

Comment: Table 1page 13

Haemoglobin phenotype -'S".

Suggestion: What is Hb 'S' phenotype? Is it homozygous sickle? Then it is better to mention 'SS' phenotype. Alignment of the variables in the table to be corrected.

Response: The term "Hb S phenotype" is used intentionally to describe the electrophoretic band observed during testing. Due to the possibility of comigrating haemoglobin’s, such as Hb S and Hb D, both of which can appear on the same band, the testing technique necessitates reporting the observed band rather than presuming the specific genotype or assuming homozygosity. This approach ensures accuracy and avoids potential misrepresentation of results. Regarding the alignment of variables in the table, we have reviewed and corrected any inconsistencies to enhance readability and ensure proper formatting.

Comment: Reticulocyte characteristics of patients with abnormal haemoglobin variantsPage 14; Lines 159- The median reticulocyte count and the absolute reticulocyte count were 0.71 x10^6/μL and 5.40%, respectively.

Suggestions: How can the absolute count be in percentage? Always mention range along with median.

Response: Authors have revised this section. See line 159-160.

Comment: Lines 161-162

The median low, medium and high fluorescence levels were 63.70%, 17.80%, and 17.50%, respectively.

Suggestion: Is it the median low, medium and high fluorescence reticulocyte %?

Response: Yes. See line 163-164.

Comment: Lines 163-164: Most 163 of the participants had elevated reticulocyte counts (60.8%), reticulocytes (76.5%), and 164 immature reticulocyte fractions (94.5%).

Suggestion: Sentence is not clear. How are Reticulocyte count and reticulocytes different here?

Response: Corrections has been revised accordingly. See line 165-167.

Comment: Lines 164-166: Furthermore, most had elevated medium (89.4%) and high (94.5%) fluorescence reticulocytes. However, most had decreased low fluorescence (92.6%) and reticulocyte haemoglobin levels (67.1%).

Suggestion: Rephrase the sentence. The message conveyed is not clear.

Response: Authors have rephrased the sentence as suggested. See line 165-167.

Comment: Table  Pages 14, 15

When you are mentioning absolute Reticulocyte count, Is there any significance for the reticulocyte %? Better index would be the reticulocyte index or reticulocyte production index.

The last portion in the table 2, can be incorporated in the upper part of the table only.

Eg;

Table 2: Reticulocyte characteristics of study participants (N=217)

Variable Median (range)/N(%)

Reticulocyte Count (106/uL) 0.17 (0.10-0.25)

Normal 16 (7.4)

Response: Comments have been accepted and revised according. See line 160-161 and table 2. The last portion of table 2 had different unit incorporating it at the upper part of the table might make reading difficult for our audience.

Comment: Sociodemographic factors associated with the severity of anaemia page 15

Lines- 177-

Additionally, most patients with mild (53.5%) and moderate (60.2%) anaemia had 'S' haemoglobin phenotypes, followed by 'SC' haemoglobin phenotypes (39.5% and 36.7%, respectively). Similarly, most subjects with severe (65.4%) and severe life-threatening (66.7%) anaemia had 'SS' haemoglobin phenotypes followed by 'SC' phenotypes (28.8% and 29.2%, respectively).

Suggestion: Is Haemoglobin 'S' and 'SS' the same? Better to use the same short forms

Response: Authors have revised this section. See line 150, 180-182.

Comment: DiscussionPage 21

Contrary to this study, (8) reported that SS and AS were significantly associated with severe anaemia. (14) in Sri Lankan adolescents also reported that haemoglobin variants are significantly associated with haemoglobin levels.

Suggestion: Please mention the study/author details; not just reference number.

Response: Authors have revised this section accordingly. See line 281-283.

Comment: Lines 297-

Interestingly, most patients with abnormal haemoglobin variants have been found to experience haemolytic episodes Patel et al., (2021) and infections Alkot et al., (2018). This leads to increased anaemia and haematopoiesis to supplement the metabolic rates of the body.

Suggestion: Sentences need rephrasing. Do not convey the message. Use same pattern of reference through out the manuscript.

Response: Authors have revised this section as suggested. See line 289-290.

Comment: The result and discussion parts are too long and confusing the reader. When all the findings are mentioned in the tables, mention only the relevant findings in the text part of the result. The sentences in most places are incomplete and do not convey the clear message.

The patient selection, characteristics, inclusion and exclusion criteria are not clearly mentioned.

Response: Though the authors do not clearly understand what the reviewer required here, we have read and address most of the issues raised with clarity.

Comment: How many were transfusion dependent/symptomatic?

Response: This was beyond the scope of our study. Thank you.

Comment: If transfusion dependent, what was the interval between transfusion and blood analysis for study in these patients?

Response: This was beyond the scope of our study. Thank you.

Comment: Was the iron study done?

Response: No please. Thank you.

Comment: Was any patient on iron chelation?

Response: No please. Thank you.

Comment: Was there splenomegaly?

Response: No please. Thank you.

Reviewer #2:

The manuscript has been well written but however, please go through my comments and make the necessary corrections suggested. I am commending you again for a good work done. I hope this continues and i get to review more papers from you.

Abstract

Comment: The abstract could be enhanced by briefly mentioning the clinical implications of the findings, particularly how reticulocyte analysis could influence patient care.

Response: Authors have revised this section. See line 48-55.

Comment: A more detailed summary of limitations would be beneficial, as this would provide a realistic perspective on the study’s generalizability.

Response: Limitation have been duly acknowledged at the limitation section. See line 359-361.

Introduction

The introduction clearly articulates the importance of the study, linking reticulocyte analysis to the management of abnormal hemoglobin variants, which is a valuable diagnostic gap. However, I recommend the following for improvements:

Comment: Key terms, such as “production defects” and “maturation defects,” are introduced but not adequately defined. Providing clear definitions upfront would enhance readability, especially for a broader audience.

Response: Authors have defined and added reference for these definitions. See line 116-120.

Comment: The introduction could be strengthened by mentioning prior studies, if any, that relate reticulocyte parameters to hemoglobinopathies. This would provide a clearer context for the study's novelty.

Response: Although there are no recent studies in regard to our study, authors have revised this section accordingly. See line 63-64

Methods

Comment: The use of convenience sampling and the limited three-month recruitment period introduce potential biases. The methodology should address the possible impact of this sampling approach on the findings.

Response: The convenience sampling, we believe, had minimal or no impact on the findings of this study because the individuals with these abnormal haemoglobin variants attended a structured clinic where they could be accessed easily hence their targeting. Also, the three months period was sufficient to obtain the appropriate sample size to give us a statistical power of analysis due to the sampling technique

Comment: The Methods section does not mention controlling for confounding factors (e.g., nutritional status, infection history) that could affect both reticulocyte indices and hemoglobin levels. Including these factors would strengthen the study’s rigor.

Response: We agree to this suggestion and has captured it as a limitation. See line 361-363.

Comment: The lack of a control group limits the ability to compare findings against a baseline. Including or discussing a control group (healthy individuals or those without abnormal hemoglobin variants) would make results more meaningful.

Response: Authors have capture it as a limitation to be considered in future studies. See line 361-363.

Results

Comment: While informative, the visuals could benefit from more descriptive labels and clearer footnotes. For example, Figure 1’s asterisks lack a corresponding explanation about their statistical significance levels.

Response: Asterisks have been defined.

Comment: The high prevalence of maturation defects (91.7%) is reported, but the clinical significance of this finding is not thoroughly discussed here. Readers would benefit from a brief comment on why this statistic is relevant for clinical settings.

Response: Clinical significance of our study findings was discussed in the discussion section, from line 299-304.

Discussion

Comment: Links to prior studies and theories (e.g., chronic hemolytic episodes and organ damage) are well integrated, supporting the study's interpretation. The following are some areas for Improvement:

Comment: The discussion should acknowledge the potential influence of unmeasured factors like nutritional status or recent blood transfusions, which could impact reticulocyte counts.

Response: Authors have revised accordingly. See line 299-304.

Comment: The manuscript would benefit from a clearer emphasis on the clinical applications of fluorescence reticulocyte measurements in patient management and treatment planning.

Response: Authors have revised accordingly. See line 323-331.

Comment: Without a control group, it is challenging to contextualize the results within a standard population. The discussion should reflect on this limitation and suggest how future studies might address this gap.

Response: Authors have revised this accordingly. See line 342-349.

Conclusion

Comment: The conclusion could benefit from more specific recommendations for clinical practice. While the potential for reticulocyte analysis is highlighted, providing explicit guidance on how these findings could alter current protocols would make the conclusion more impactful.

Response: Suggestions have been accepted and revised accordingly. See line 351-358.

Comment: A broader reflection on the study’s limitations and suggestions for future research, especially the need for multicenter studies, would also improve the depth of this section.

Response: Suggestions accepted and revised accordingly. See line 361-363.

Limitations and Recommendations

Comment: Additional discussion on sampling limitations (e.g., convenience sampling) and their impact on data generalizability would provide readers with a fuller perspective on the study's constraints.

Response: Suggestions accepted and revised accordingly. See line 361-364.

Comment: The recommendations are valid but could be enhanced by proposing specific avenues for future research, such as exploring the role of reticulocyte indices in therapeutic monitoring for hemoglobinopathies.

Response: Suggestions accepted and revised accordingly. See line 368-372.

Language and Technical Presentation

Comment: A few spelling errors (e.g., “recticulocytes” instead of “reticulocytes”) and minor grammatical mistakes were found, which could detract from the professionalism of the manuscript. Careful proofreading would resolve these.

Response: Authors have proofread manuscript again and corrected minor spelling errors.

Comment: Consistently defining and using terms throughout would improve readability, especially for an international audience not familiar with specific local practices.

Response: Authors have accepted suggested corrections and revised accordingly.

Reviewer #3:

Comment: Abstract contains repeated statements which will need to be corrected.

Response: Could the reviewer be specific??

Comment: Conclusion in abstract and the conclusion section of the manuscript should be re-written to clearly state findings and implications without necessarily repeating the results segment.

Response: Authors have revised accordingly at conclusion in abstract and the conclusion section of manuscript. See line 48-55 in abstract and line 349-356.

Comment: Statement on written consent and assent was repeated under sample collection and ethical approval. Several other repeated statements in the manuscript should be avoided.

Response: Authors have revised the written consent accordingly. See line 144-146.

Comment: "The median age of the study participants was 7.00 years, with the majority within 1-5 years (40.1%), followed by 6-10 years (29.0%). Half were men (50.2%) and half were women (49.8%)." - It is better to use males and females especially as the mean age of participants was just 7 years

Response: Comment has been revised accordingly. See line 149-150.

Comment: The levels of HB used to define and categorize anaemia should be stated either be in text or in the table especially since the age range of the study participants cut across both paediatric and adult age groups.

Response: Comments has been revised accordingly. See page 122-125.

Comment: Figures presented do not adequately explain the findings described in text for example, the association between reticulocyte production indices and haemoglobin level and the association between reticulocyte maturation-dependent indices and haemoglobin levels among patients with abnormal haemoglobin variants would have been better presented in tables.

Response: We acknowledge the comment to present the associations between reticulocyte production indices and haemoglobin levels, as well as reticulocyte maturation-dependent indices and haemoglobin levels, in tables. However, we believe that the figures included in the manuscript are the most effective way to convey these associations. The figures provide a clear visual representation of the relationships, enabling readers to grasp trends and patterns at a glance. These visualizations effectively highlight the strength and nature of the associations, which may be less apparent in a tabular format. Therefore, we respectfully believe that the figures best serve the purpose of illustrating these findings and align with the manuscript's goals of clarity and impact. Authors have explained findings in text at the result section.

Comment: Referencing in the discussion section does not follow acceptable methods. Some referenced authors were not named in text while some that were named did not have the reference number indicated. (Line 289, 290 and 298 are examples)

Response: Authors have revised the suggested comments accordingly.

Comment: The authors did not explain why t

---

## [Decision Letter · Decision Letter 1]

28 Feb 2025

PONE-D-24-30332R1Unlocking the Power of Reticulocyte Analysis: Advancing care for patients with abnormal haemoglobin variants at Eastern Regional Hospital, KoforiduaPLOS ONE

Dear Dr. Duneeh,

Thank you for submitting your manuscript to PLOS ONE. After careful consideration, we feel that it has merit but does not fully meet PLOS ONE’s publication criteria as it currently stands. Therefore, we invite you to submit a revised version of the manuscript that addresses the points raised during the review process.

**Overall the manuscript has merit and meets the standards for the publication but the abstract part is too long (391 words) it is hard to follow. Please shorten it. And the language with the abstract part is somehow challenging. And some parts of the abstract is written and generated by AI. Please remove the parts that was generated by AI and re-write the abstract. It didn't make sense to write "this study" when reporting your own manuscirpt. Please change or correct it.**

We look forward to receiving your revised manuscript.

Kind regards,

Mehmet Baysal

Academic Editor

PLOS ONE

**Journal Requirements:**

Reviewers' comments:

Reviewer's Responses to Questions

**Comments to the Author**

1. If the authors have adequately addressed your comments raised in a previous round of review and you feel that this manuscript is now acceptable for publication, you may indicate that here to bypass the “Comments to the Author” section, enter your conflict of interest statement in the “Confidential to Editor” section, and submit your "Accept" recommendation.

Reviewer #1: All comments have been addressed

Reviewer #2: All comments have been addressed

Reviewer #3: All comments have been addressed

2. Is the manuscript technically sound, and do the data support the conclusions?

Reviewer #1: Yes

Reviewer #2: Yes

Reviewer #3: Yes

3. Has the statistical analysis been performed appropriately and rigorously? 

Reviewer #1: Yes

Reviewer #2: Yes

Reviewer #3: Yes

4. Have the authors made all data underlying the findings in their manuscript fully available?

Reviewer #1: Yes

Reviewer #2: Yes

Reviewer #3: Yes

5. Is the manuscript presented in an intelligible fashion and written in standard English?

Reviewer #1: Yes

Reviewer #2: Yes

Reviewer #3: Yes

6. Review Comments to the Author

**Reviewer #1:**  This is an interesting study with potential impact on clinical practice.

**Reviewer #2:**  All comments reserved as authors have addressed all comments. Authors have adequately addressed my comments raised in a previous round of review.

**Reviewer #3:**  Issues highlighted in the first review have been addressed by the authors. I am satisfied with the corrections implemented and the current state of the manuscript.

7. PLOS authors have the option to publish the peer review history of their article (what does this mean? ). If published, this will include your full peer review and any attached files.

**Do you want your identity to be public for this peer review?** For information about this choice, including consent withdrawal, please see our Privacy Policy .

Reviewer #1: No

Reviewer #2: **Yes: ** Juliana Aggrey

Reviewer #3: No

---

## [Author Response · Author response to Decision Letter 2]

27 Mar 2025

Journal Requirements

Comment: Please review your reference list to ensure that it is complete and correct. If you have cited papers that have been retracted, please include the rationale for doing so in the manuscript text, or remove these references and replace them with relevant current references. Any changes to the reference list should be mentioned in the rebuttal letter that accompanies your revised manuscript. If you need to cite a retracted article, indicate the article’s retracted status in the References list and also include a citation and full reference for the retraction notice.

Response: References were checked for completeness. No changes have been made. Thank you

Comment: “Overall the manuscript has merit and meets the standards for the publication but the abstract part is too long (391 words) it is hard to follow. Please shorten it. And the language with the abstract part is somehow challenging. And some parts of the abstract is written and generated by AI. Please remove the parts that was generated by AI and re-write the abstract. It didn't make sense to write "this study" when reporting your own manuscirpt. Please change or correct it.”

Response: The manuscript abstract has been revised manually to reflect editors suggestions and comments. See line 21-44.

Reviewer #1: This is an interesting study with potential impact on clinical practice.

Response: Thank you for dedicating your time to reviewing our manuscript. We, the authors, greatly appreciate your expertise and insights in enhancing the quality of our work..

Reviewer #2: All comments reserved as authors have addressed all comments. Authors have adequately addressed my comments raised in a previous round of review.

Response: Thank you for dedicating your time to reviewing our manuscript. We, the authors, greatly appreciate your expertise and insights in enhancing the quality of our work.

Reviewer #3: Issues highlighted in the first review have been addressed by the authors. I am satisfied with the corrections implemented and the current state of the manuscript.

Response: Thank you for dedicating your time to reviewing our manuscript. We, the authors, greatly appreciate your expertise and insights in enhancing the quality of our work.

---

## [Editor Report · Decision Letter 2]

1 Apr 2025

Unlocking the Power of Reticulocyte Analysis: Advancing care for patients with abnormal haemoglobin variants at Eastern Regional Hospital, Koforidua

PONE-D-24-30332R2

Dear Dr. Duneeh,

We’re pleased to inform you that your manuscript has been judged scientifically suitable for publication and will be formally accepted for publication once it meets all outstanding technical requirements.

Kind regards,

Mehmet Baysal

Academic Editor

PLOS ONE
---

## [Editor Report · Acceptance letter]

PONE-D-24-30332R2

PLOS ONE

Dear Dr. Duneeh,

I'm pleased to inform you that your manuscript has been deemed suitable for publication in PLOS ONE. Congratulations! Your manuscript is now being handed over to our production team.

Kind regards,

on behalf of

Dr. Mehmet Baysal

Academic Editor

PLOS ONE